# Teichoic acids anchor distinct cell wall lamellae in an apically growing bacterium

Eveline Ultee [1,2], Lizah T. van der Aart [1,2], Le Zhang[1,2], Dino van Dissel [1,2], Christoph A. Diebolder[3], Gilles P. van Wezel [1,2], Dennis Claessen [1,2] & Ariane Briegel [1,2 ✉]

The bacterial cell wall is a multicomponent structure that provides structural support and protection. In monoderm species, the cell wall is made up predominantly of peptidoglycan, teichoic acids and capsular glycans. Filamentous monoderm Actinobacteria incorporate new cell-wall material at their tips. Here we use cryo-electron tomography to reveal the architecture of the actinobacterial cell wall of *Streptomyces coelicolor*. Our data shows a density difference between the apex and subapical regions. Removal of teichoic acids results in a patchy cell wall and distinct lamellae. Knock-down of *tagO* expression using CRISPR-dCas9 interference leads to growth retardation, presumably because build-in of teichoic acids had become rate-limiting. Absence of extracellular glycans produced by MatAB and CslA proteins results in a thinner wall lacking lamellae and patches. We propose that the *Streptomyces* cell wall is composed of layers of peptidoglycan and extracellular polymers that are structurally supported by teichoic acids.

[1] Department of Molecular Biotechnology, Institute of Biology, Leiden University, Sylviusweg 72, 2333 BE Leiden, The Netherlands. [2] Centre for Microbial Cell Biology, Leiden University, Leiden, The Netherlands. [3] Netherlands Centre for Electron Nanoscopy (NeCEN), Einsteinweg 55, 2333 CC Leiden, The Netherlands. ✉email: a.briegel@biology.leidenuniv.nl

Bacteria are successful organisms that thrive in most environments. They withstand challenging conditions by synthesizing a stress-bearing cell wall, which provides rigidity to cells and defines their shape. As the cell wall is in most cases essential for the cells' survival it is a prime target for antibiotic treatment. The core component of the cell wall is formed by a mesh of N-acetylglucosamine (GlcNAc) and N-acetylmuramic acid (MurNAc) glycan chains, which are cross-linked by peptide stems to form the peptidoglycan (PG) sacculus[1]. Bacteria have been classified into two groups based on their cell-envelope architecture: monoderm bacteria have a cell envelope that consists of a cytoplasmic membrane and a multilayered envelope, consisting of PG, teichoic acids (TAs)[2,3] and a variety of capsular glycans[4–7]. Teichoic acids are long, anionic polymers and can be classified into wall teichoic acids (WTAs) and lipoteichoic acids (LTAs), based on their linkages to either the PG or lipid membrane, respectively[8,9]. They contain similar repetitive units linked by phosphodiester linkages, which make up the long structure and render the cell envelope negatively charged[9,10]. In contrast, diderm bacteria have a thin PG layer that is positioned between a cytoplasmic membrane and an additional outer membrane containing lipopolysaccharides (LPS)[11].

In order for a bacterial cell to grow, the PG needs to expand and be remodeled by the insertion of newly synthesized PG strands into the pre-existing sacculus. The cell wall can expand either by lateral insertion of new cell-wall material enabling lateral elongation of the cell or insertion at the tip resulting in apical growth (Fig. 1a). In many unicellular rod-shaped bacteria, such as in the model organisms Escherichia coli and Bacillus subtilis, the cell wall expands by the incorporation of new PG along the length of the cell[12,13]. This lateral elongation is guided by highly curved MreB filaments along the cell circumference[14–17]. In contrast, other bacteria, including actinobacteria and Agrobacterium species, grow from their cell pole, which is independent of MreB[18,19]. In actinobacteria, the curvature sensitive protein DivIVA localizes to the cell poles and recruits the cell-wall synthesis machinery, consisting of penicillin-binding proteins and RodA[20–22]. The current model of tip growth involves de novo PG synthesis at the apex guided by DivIVA and modification of the new wall material by L-,D-transpeptidases along the lateral wall[23–26].

To address the organization of the cell-envelope architecture of polar growing bacteria, we studied the model organism Streptomyces coelicolor[27]. Streptomyces species are soil-dwelling and multicellular actinobacteria that form long, branched, hyphal cells, which collectively form an extended mycelial network[28,29]. When grown in liquid, the hyphae aggregates into dense mycelial pellets by producing a glue-like extracellular matrix (Fig. 1b)[30]. This extracellular matrix is composed of a large variety of polymers[4,31,32]. Two of these polymers have been characterized as key elements for pellet formation: one is poly-β-1,6-N-acetylglucosamine (PNAG), produced by the MatAB proteins[33,34]. The other one is a cellulose-like glycan, formed by the cellulose synthase-like enzyme CslA, which operates in conjunction with GlxA and DtpA at the hyphal tips[35–38]. They were shown to add to the stability of the tip and are important for invasive growth[35,39]. Deletion of either the matAB cluster (SCO2963 and SCO2962)[33,34] or cslA (SCO2836)[35] results in dispersed growth, which is characterized by the absence of pellet formation in liquid growth media. As the tips of growing hyphae continuously incorporate nascent PG, the architecture of the sacculus is likely structurally distinct from the mature cell wall at the lateral sides of the hyphae. Here we reveal structural differences of the cell wall at both the lateral sides and the nascent cell wall at the hyphal tip by applying cryo-electron tomography (cryo-ET) using a Volta Phase Plate (VPP) to allow high-resolution imaging with high contrast.

The results of this study provide insight into the architecture of the cell wall of a polar growing bacterium. The data suggest that the cell wall of S. coelicolor comprises two distinct lamellae. The inner lamella is composed of PG, and the glycans produced by the CslA and MatAB proteins form a discrete outer lamella, which is tethered to the inner layer via teichoic acids. Collectively, these findings warrant a revised model for the cell-wall architecture in a polar growing bacterium.

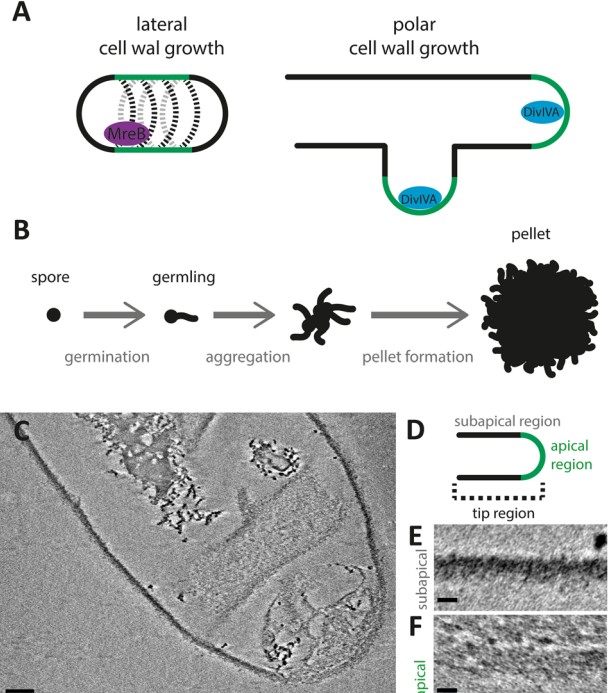

**Fig. 1 Model of *Streptomyces* growth and cryo-ETs of isolated *S. coelicolor* sacculi.** The cell wall of bacteria is incorporated either at the lateral sides via the MreB filaments or at the apex driven by DivIVA. These processes are termed lateral and polar cell-wall growth, respectively (**a**). The growth and development of *Streptomyces coelicolor* spores in liquid medium, where the spores germinate, aggregate and form dense pellets facilitated by an extracellular matrix (**b**). **c** A cryo-electron tomography slice of the tip region of a *S. coelicolor* sacculus with insets (*XY* plane) displaying differences in cell-wall structure in the apical (**e**) and subapical regions (**f**), as depicted in the schematic (**d**). Scale bar **c** 100 nm, scale bars insets **e**, **f** 20 nm.

## Results

**Isolated sacculi reveal density differences in hyphal tips**. In order to reduce the thickness of the sample and achieve high-resolution cryo-electron tomograms of the cell envelope of S. coelicolor, we chemically isolated sacculi[40,41]. We then used cryo-ET in combination with a Volta phase plate (VPP) to increase the low frequency contrast of our data[42].

Tomograms of some of the sacculi displayed extensive folding perpendicular to the long axis of the hyphae. These folds are similar in orientation compared to the shears and tears previously observed in sacculi preps of B. subtilis[43], likewise supporting a circumferential orientation of the glycan strands in the cell wall (Supplementary Fig. 1)[43]. The cell wall of the purified WT sacculi has a thickness of ~30 nm, similar to the width of the cell wall of B. subtilis sacculi[43]. Furthermore, the tomograms of the S. coelicolor sacculi showed that the apex of the tip region appears less densely packed with cell-wall material (Fig. 1c, d). The subapical parts (Fig. 1e) exhibit a higher contrast compared to the apical parts (Fig. 1f). Of the nine datasets we have recorded, two

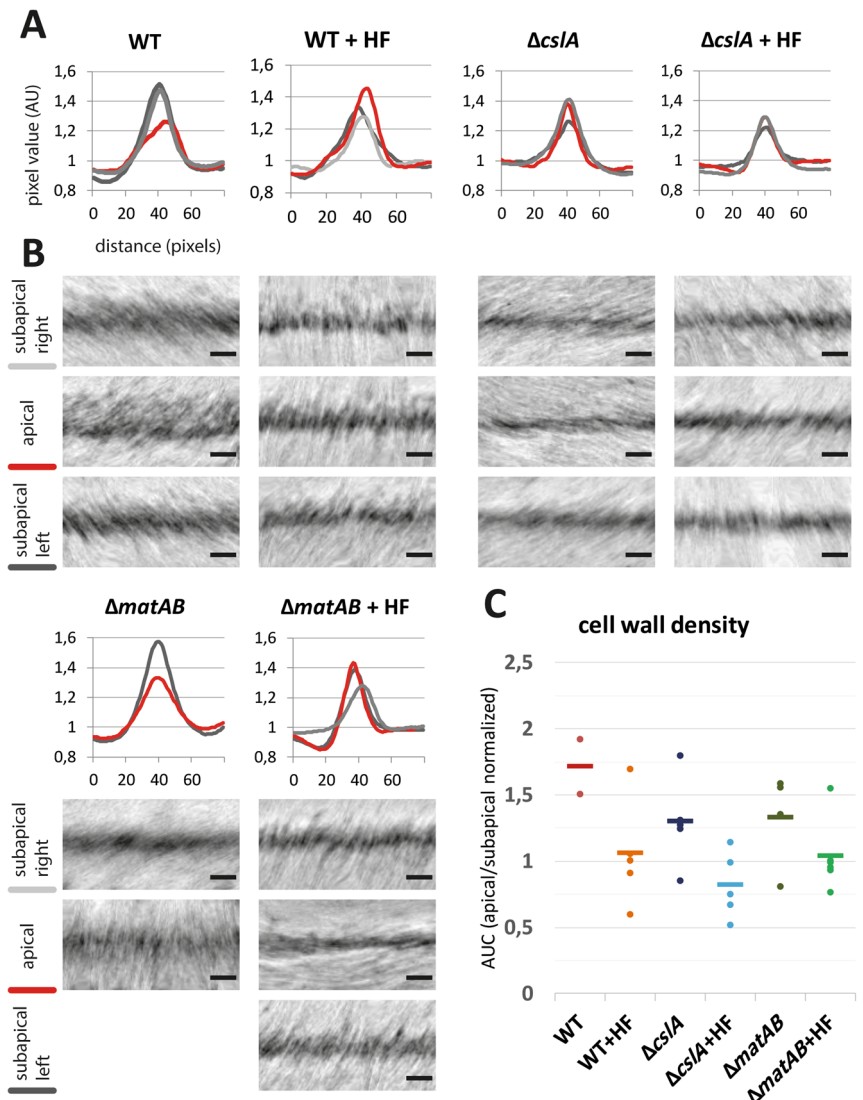

**Fig. 2 Density profiles of the apical and subapical regions of isolated sacculi.** Cryo-electron tomographic slices of *S. coelicolor* WT, *ΔcslA* and *ΔmatAB*, sacculi, either treated with HF and non-treated, were used to analyze the density of the cell wall. For each sacculus, the apical region (red) together with the subapical regions on the left (dark gray) and right side (light gray) were segmented and straightened in the *XY* plane. The density was determined by averaging pixel values (arbitrary units) along the straightened cell-wall regions, and normalized against the background pixel values. These pixel values are presented in the density plots relative to the distance perpendicular to the cell wall (**a**). Representable cryo-ET fractions of the segmented and straightened cell walls in the *XY* plane are depicted in the panels below the density plot of each strain (**b**). Not all sacculi contain a left and right subapical region in the tomogram field of view, as depicted by the missing panel in the *ΔmatAB*. For each sacculus, the area under the curve (AUC) of the density plot was determined, at which the AUC of the apical region was normalized with the AUC of subapical region (left and right averaged, if both available) and plotted in **c**. Hence, the value of 1 indicates that the apical and subapical region are comparable in terms of cell-wall density, whereas a value deviating from 1 indicates a denser (<1) or less dense (>1) apical cell-wall region. The normalized AUC of each sacculus is depicted as a single dot, the horizontal line depicts the group average (WT $n = 2$, WT + HF $n = 5$, *ΔcslA* $n = 5$, *ΔcslA* + HF $n = 5$, *ΔmatAB* $n = 4$, *ΔmatAB* + HF $n = 6$).

tips indeed showed a striking difference in electron density between the apical and subapical regions (Fig. 2a). In the other tips, the apical regions were not conspicuously different compared to the subapical part. Four other tips were too closely located to the carbon support film and could not be further analyzed. This dataset indicates a structural difference between the apexes, where the incorporation of new PG into the existing cell wall takes place, and the relatively older cell wall located subapically.

**Teichoic acids provide structural support to growing tips.** To study the contribution of TAs to the overall cell-wall architecture, we treated sacculi with hydrofluoric acid (HF). HF cleaves

phosphodiester bonds and is an established method to remove TAs from the cell wall[9,44,45]. Tomograms of HF-treated sacculi showed no apparent density differences between apical and subapical regions (Fig. 2a, b). To compare the difference in electron density between apical and subapical regions of HF-treated and non-treated sacculi, the overall density was calculated (Area Under the Curve of the density plot, normalized apex by subapex) and expressed as a ratio, where 1 indicates that the apical and subapical region are comparable in terms of electron density, whereas a value deviating from 1 indicates a denser (<1) or less dense (>1) apical cell-wall region (Fig. 2c). Creating large datasets with cryo-ET experiments is challenging, as such the observations made could not be statistically substantiated due the small sampling size of our data. The data presented here indicate

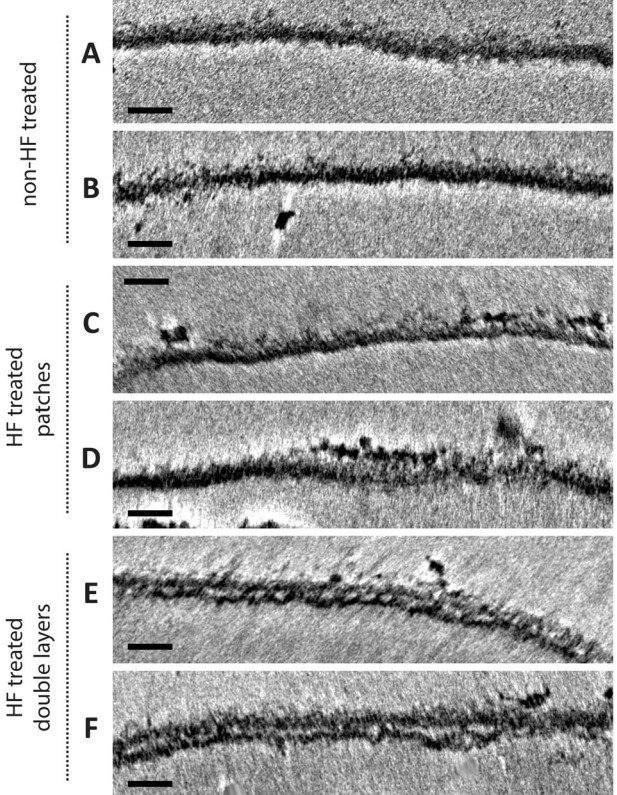

**Fig. 3 Cryo-ETs of *S. coelicolor* WT sacculi show alterations in the PG layer upon HF treatment. a**, **b** A representative part of the PG layer of a non-HF-treated *S. coelicolor* WT sacculus. **c**–**f** Representative parts of the PG layers of HF-treated sacculi, classified as patches (**c**, **d**) and double layers (**e**, **f**). All scale bars are 50 nm.

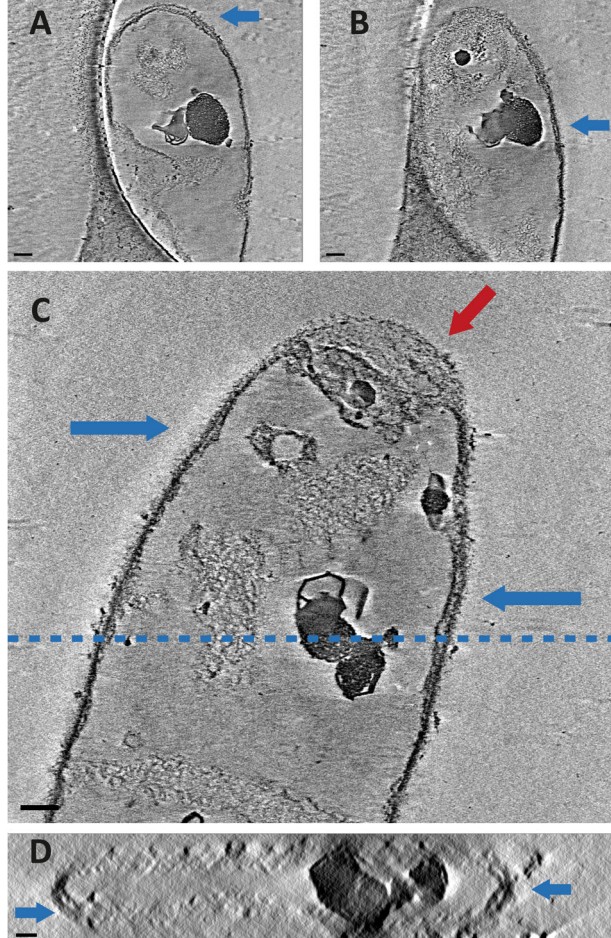

**Fig. 4 Cryo-ETs of HF-treated *S. coelicolor* WT sacculi show double layers of PG.** The micrographs in **a** and **b** are the same sacculus, displayed at different *Z* heights, the blue arrows highlight the presence of a double layer. The micrographs at **c** show a collapsed double layer at the apex, highlighted by the red arrow. The double layer of the sacculus in **c** can also be seen in a *ZX* plane (**d**) at the approximate *Y* height depicted in **c** by the dashed line. Scale bars 100 nm .

that HF treatment affects the cell-wall density differences observed between the apical and subapical regions, therefore it could be hypothesized that TAs might be involved in the cell-wall integrity.

Analysis of the HF-treated sacculi revealed another difference compared to the non-treated samples: in the absence of TAs, the structure of the sacculi was overall less uniform and revealed a patchy and rough cell-wall structure (Fig. 3). This frayed appearance of the cell wall was evident in all regions of the sacculus and not restricted to a specific area. In 3 out of 12 HF-treated sacculi, the sacculus consisted of 2 distinct lamellae (Fig. 4). The other 9 HF-treated sacculi did not reveal distinct layers but revealed a more 'patchy' cell-wall structure. The patchy cell-wall pattern was also evident in a tomogram that we collected from an emerging side branch (Supplementary Fig. 2), suggesting that it correlates to young hyphae. Taken together, this suggests that teichoic acids have a role in the structural integrity of the *S. coelicolor* cell wall.

Next, a genetic approach was used to confirm whether the structural differences revealed by the HF treatment were indeed resulting from the complete removal of the teichoic acids. For this, we attempted to generate a knock-out mutant lacking *tagO*, which encodes the enzyme catalysing the first step in teichoic acid synthesis. The TagO protein transfers GlcNAc-1-P from UDP-GlcNAc to a membrane-anchored UDP carrier lipid. This carrier lipid is shared with the PG biosynthesis pathway, whereas steps thereafter are committed to teichoic acid synthesis specific components[8,46]. We attempted to create a *tagO* null mutant in *S. coelicolor* by replacing the *tagO* gene (SCO5365) with the apramycin resistance cassette *aac(C) IV* using the unstable multi-copy vector

pWHM3, as described[47]. Despite many attempts, this did not yield viable transformants. This is in line with earlier experiments (G. Muth, pers. comm.). As an alternative, we used CRISPR interference by targeting the catalytic dead dCas9 protein to the *tagO* gene in *S. coelicolor* M145[48]. This allows investigation of the effect of strongly reduced expression of a gene of interest.

The dcas9 and sgRNA scaffolds were expressed from the strong and constitutive *gapdhp* and *ermEp* promoters, respectively, so that the dCas9:sgRNA complex was continuously expressed at a high level, and therefore did not require induction. When the dCas9:sgRNA complex binds to the template strand, the sgRNA faces the RNA polymerase (RNAP) and may be replaced by the helicase activity of the RNAP, allowing transcription to continue undisturbed[49]. Conversely, when the dCas9:sgRNA complex binds to the non-template strand, the dCas9:sgRNA will face the RNAP, which cannot be replaced, resulting in transcriptional pausing[49]. Making use of these properties, knockdown of *tagO* was enforced from the CRISPRi construct pGWS1365 that expresses a spacer targeting the non-template strand of *tagO*. Expectedly, growth was strongly inhibited in the *tagO* knockdown strain, as teichoic acids are essential for growth. Such growth inhibition was not seen in the control strain harboring a construct that targets the template strand (Supplementary Fig. 3A).

However, some transcription of *tagO* may occur, as a result of the gradual depletion of the antibiotic that is required for plasmid maintenance, which will result in partial loss of the plasmid. This will then allow the production of a small pool of teichoic acids, enabling growth. These hyphae have a wild-type cell-wall composition, as seen by our experiments.

The delayed growth of the *tagO* knockdown strain as well as impaired spore germination in submerged cultures became more apparent upon sacculi isolation. From the control strain, the individual hyphal tips could clearly be examined as they protruded from the mycelium, whereas sacculi isolation of the *tagO* mutant resulted in large clumps with barely any hyphae distinguishable from the high amount of non-germinated and aggregated spores (Supplementary Fig. 3B). Although no further high-resolution cryo-ET on sacculi of the *tagO* mutant was feasible, cryo-TEM images of mycelia showed that there were no severe aberrations in cell-envelope morphology or thickness between the hyphal tips of the tagO knockdown and the control strain (Supplementary Fig. 3C). Although in-depth structural analysis on the tagO mutant cell wall could not be performed, the results presented here indicate that the TAs themselves do not form an additional layer or add to the thickness of the cell wall.

**Extracellular glycans form a distinct second lamella**. To investigate whether the patches and double layers are composed of PG, HF-treated sacculi were treated with mutanolysin that cleaves β-*N*-acetylmuramyl-(1→4)-*N*-acetylglucosamine linkages in PG[50,51]. Exposure to mutanolysin indeed degraded most of the sacculi within 15 min and after 30 min most individual sacculi were absent (Supplementary Fig. 4). This confirms that the PG is the shape-determining component of the sacculus but does not yet provide an insight into the causes of the patches and distinct layers observed with cryo-ET.

When grown in liquid media, *S. coelicolor* germlings produce an extracellular matrix leading to self-aggregation into dense pellets[52]. This pellet structure remains intact after chemical isolation of the sacculi and even after the mutanolysin treatment as observed by light microscopy (Supplementary Fig. 4). The extracellular matrix is composed of a large variety of glycans, such as poly-*N*-acetylglucosamine (PNAG) and a cellulose-like polymer, produced by MatAB and by CslA, respectively[33–35]. As the pellet structure is preserved even after sacculi isolation, these polymers might have a direct role in the cell-wall architecture of *S. coelicolor*.

To determine the role of these extracellular glycans in the cell-wall architecture of *S. coelicolor*, we isolated the sacculi of two mutants' strains: *S. coelicolor* M145 Δ*matAB* and Δ*cslA*. As expected, both mutant strains have a clear open morphology compared to the dense pellets formed by the wild-type (Supplementary Fig. 5). The data acquired from the *matAB* and *cslA* mutants appeared similar to the sacculi of the non-HF-treated WT at first glance (Fig. 5a). The hyphal tips did not show a clear difference between the electron densities at the apex, compared with the lateral or subapical regions as seen in the WT (Fig. 2a, b). However, in contrast to the non-treated WT, both the non-treated Δ*cslA* and Δ*matAB* sacculi sporadically revealed two lamella similar as seen in the HF-treated WT. This could indicate that not merely the TAs, but also extracellular polymers are required for the integrity of the cell wall. In data acquired of HF-treated sacculi of both *matAB* and *cslA* mutants on the other hand, we did not observe patches and double layers in either mutant (Fig. 5).

Furthermore, comparison of these mutants with the parental strain revealed difference in cell-wall thickness (Fig. 5b). WT sacculi had an average thickness of 30.16 nm (±0.88, *n* = 9). The

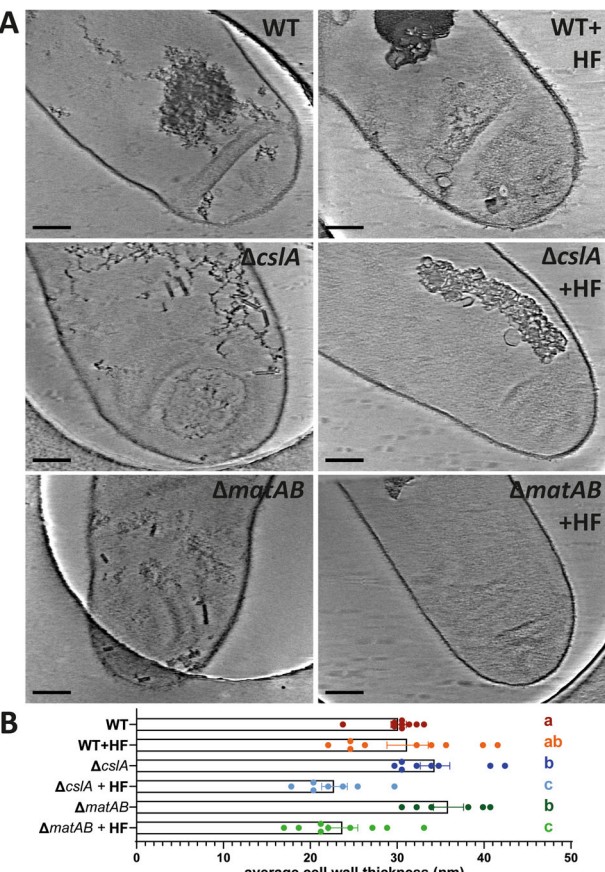

**Fig. 5 HF-treated sacculi of *S. coelicolor* Δ*matAB* and Δ*csla* appear thinner.** The micrographs show sacculi of *S. coelicolor* WT, Δ*cslA* and Δ*matAB* strains either treated with HF or non-treated (**a**), scale bar 200 nm. The thickness of the overall saccculi are measured and shown in the graph (**b**). The error bars depict the standard error of the mean. Different letters denote the statistically significant differences among the samples, at a *p*-value of <0.05 following the Students *t*-test (*n* = 9 individual sacculi measured for WT, WT + HF, Δ*matAB* + HF, *n* = 8 for Δ*cslA*, *n* = 7 for Δ*cslA* + HF, *n* = 6 for Δ*matAB*). The samples with the same letter have no statistically significant difference from each other.

thickness of HF-treated WT sacculi showed an average thickness of 31.19 nm (±2.37, *n* = 9) and a higher variance, presumably caused by the patches and double layers in this sample group. In contrast, the non-treated sacculi of the Δ*matAB* and Δ*cslA* strains showed an average thickness of 35.91 nm (±1.73, *n* = 6) and 34.35 nm (± 1.71, *n* = 8), respectively. Although HF treatment leads to a thinner cell wall of 23.75 nm (±1.72, n = 9) for Δ*matAB* and 22.78 nm (±1.49, *n* = 7) for the Δ*cslA* mutant strain. The values of the parental strain significantly differed from the cell-wall thickness of the non-treated sacculi of *matAB* and *cslA* mutants (*p* < 0.05) and of those of HF-treated mutant sacculi (*p* < 0.005) when analyzed using a Student's *t*-test.

The remarkably thicker cell wall and sporadic appearance of cell-wall lamella in sacculi of both the *matAB* and *cslA* null mutants suggests that the absence of either extracellular polymer negatively influences the integrity and compact nature of the cell wall. In addition, the thinner cell wall of both the *matAB* and *cslA* null mutants after HF treatment indicates that the extracellular glycans produced by the MatAB proteins and CslA comprise a considerable part of the cell wall itself. Moreover, the absence of patches and double layers in the HF-treated mutants imply that TAs to a large degree have a role in the anchoring of patches and

layers of glycans to the cell wall. This suggests that the glycans together with the TAs are an integral part of the cell wall.

## Discussion

In this work, we show that the cell envelope of the polar growing bacterium *S. coelicolor* is a complex structure composed of PG and extracellular glycans that are structurally linked by teichoic acids. Our observations indicate that the TAs directly add to the structural integrity of the *S. coelicolor* cell wall. Chemical removal of the TAs reveals the existence of two distinct lamella, or likely remnants thereof, appearing as a patchy cell wall. These observations were not restricted to the apical region and were also seen subapically. The structural integrity of the cell wall was additionally affected in sacculi of *cslA* and *matAB* null mutants, lacking the extracellular polymers. The absence of extracellular polymers led to sacculi with a thicker appearance, with sporadically revealing lamella comparable to those observed in HF-treated WT sacculi. In contrast, the HF-treated sacculi of *cslA* and *matAB* null mutants lacked any observable lamellae, strongly suggesting that lamellae are composed of—or depend on—the extracellular glycans synthesized by the CslA and MatAB proteins. Based on our work, we propose a revisited model for the cell wall of the polar growing bacterium *S. coelicolor* (Fig. 6).

Cryo-ET has been used previously to study for example cell division events in *Streptomyces*[53]. However, to our knowledge this is the first study investigating the cell-wall structure in this organism. Our results align well with previous Cryo-ET work in respect to the orientation of the glycan strands in the sacculus. In these studies, the glycan strand orientation in the PG of the laterally growing diderm bacteria *E. coli* and *Caulobacter crescentus*[54] and the monoderm *B. subtilis*[43] were shown to be circumferentially positioned around the bacterium. None of these studies reported a difference between the apical and lateral parts of the sacculi. In contrast, we show here that the apical regions appear less densely packed compared to the lateral regions in *S. coelicolor*. However, this difference was not found in all apical regions, which is likely due to the fact that not all tips are actively growing. In addition, this difference is also not apparent in the HF-treated sacculi, which supports a growth model where the TAs may have an important role.

Our data further indicate that TAs have an essential role in the structural integrity of the cell wall by tethering the distinct lamellae of the cell wall to one another. The role of the TAs in the cell-wall structure has been previously studied in the Gram-positive pathogens *Staphylococcus aureus* and *Listeria monocytogenes*[55]. Both pathogenic bacteria showed alterations in the PG layer as a result of treatment with the antibiotic tunicamycin, which inhibits the WTA biosynthetic enzyme TagO[56]. Classical negative stain and sectioning TEM showed that the PG layer of *L. monocytogenes* and *S. aureus* appears thinner and rougher upon tunicamycin administration[55]. The micrographs from these studies show a striking resemblance with our data of the *S. coelicolor* sacculi treated with HF. In addition, early cryo-EM research on frozen-hydrated sections of *B. subtilis* cell-wall fragments also showed that removal of the TAs lead to a loss of rigidity and reduced thickness by around 10 nm[57]. This finding correlates well with the loss of integrity, we observed upon removal of TAs from the *S. coelicolor* cell wall. Reduced expression of *tagO* using CRISPR-dCas9 interference technology resulted in impaired spore germination and slow growth, obstructing proper sacculi isolation for cryo-ET analysis. The cell-wall thickness of hyphal tips was not affected by the lack of TAs, which is consistent with the perpendicular orientation of TAs to the wall[9]. We hypothesize that the reduced expression of *tagO* results in a reduced supply of TA precursors, leading to a growth delay because building the TA layer of the cell wall becomes rate-limiting. This also suggests that incorporation of TA into the cell wall is essential in *S. coelicolor*, which is consistent with our failure to create *tagO* null mutants, and with the sick appearance of the *tagO* knockdown mutants. The frayed appearance of the cell wall as observed in the HF-treated sacculi could not be detected in the *tagO* mutant, as the isolated sacculi of the mutant were unsuitable for high-resolution cryo-ET experiments.

The data presented in this study show that the *S. coelicolor* sacculus is thinner in the absence of TAs and the glycans produced by the CslA and MatAB proteins. This could indicate that both glycans form an additional rigid layer providing protection during apical growth. Both glycans are important for pellet formation in *Streptomyces* and are associated with biofilm formation in other bacteria[58–60].

In summary, the results presented here indicate that the cell wall of polar growing bacterium *S. coelicolor* contains the structurally important components PG, TAs and extracellular glycans that together compose a thick and complex cell wall. In our model, we propose that the cell wall is composed of a layer of densely cross-linked PG, with a layer of extracellular glycans produced by *cslA* and *matAB*-encoded proteins on top and exposed to the exterior of the cell. These layers are packed together by wall TAs, which are interweaved throughout the cell wall in hyphal tip. These findings lead to the insight that the *S. coelicolor* cell envelope is a complex network composed of PG and extracellular glycans, and that is structurally interlinked by TAs.

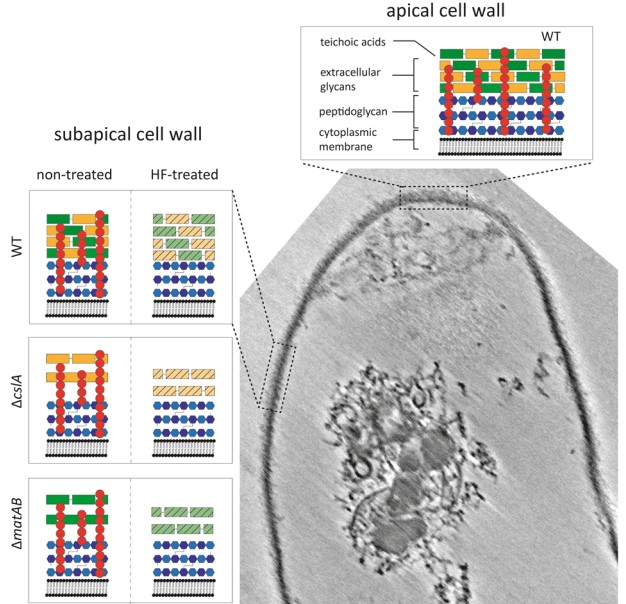

**Fig. 6 Schematic model of the cell-wall architecture of *S. coelicolor*.** Micrograph of a WT *S. coelicolor* sacculus, with insets showing the schematic model of the cell-wall architecture resulting from the data presented in this study. The model shows the cellular lipid bilayer or membrane, covered by a layer of peptidoglycan (PG) and extracellular glycans, which are interconnected by the wall teichoic acids (TAs). At the top, the apical cell-wall panel represents the apex, where new PG is incorporated and cross-linked into the sacculus. The left panels show the subapical cell wall of the WT and mutant strains, either treated with hydrofluoric acid (HF) or non-treated. Extracellular polymers produced by CslA (green) and MatAB (yellow) are absent in the mutant strains ΔcslA and ΔmatAB, respectively. The HF treatment causes the removal of the wall teichoic acids, and potentially affects the cell-wall integrity and detachment of extracellular glycans (striped). The Gram-positive cell wall is composed of the combined structure of wall teichoic acids, extracellular polymers and peptidoglycan.

**Table 1 Cryo-EM data collection, refinement and validation statistics.**

| | Sacculi tomography |
|---|---|
| Data collection and processing[a] | |
| Magnification | SA ×33,000 |
| Voltage (kV) | 300 |
| Electron exposure (e⁻/Å²) | 100 (cumulative exposure over tilt range) |
| Defocus range (μm) | 0.5 |
| Pixel size (Å) | 4.241 |

[a]Further processing of the cryo-EM data was performed as mentioned in the Methods section. No refinement and validations statistics are applicable for cryo-electron tomography data.

## Methods

**Strains.** *Streptomyces coelicolor* A3(2) M145 was obtained from the John Innes Center Strain Collection. The deletion mutants of *cslA* (SCO2836) and *matAB* (SCO2963 and SCO2962) in *S. coelicolor* M145 used in this study have been previously published by Xu et al.[35] and Van Dissel et al.[33,34], respectively. All techniques and media used to culture *Streptomyces* are described in ref. [61]. Spores of *Streptomyces* were harvested from Soy Flour Mannitol (SFM) agar plates. Fresh spores were used to inoculate 400 mL of Tryptic Soy Broth supplemented with 10% (w/v) sucrose, in 2 L flasks with coiled coils. The liquid cultures were grown while shaking at 200 rpm and 30 °C for 12 h prior to sacculus isolation.

**Sacculus isolation.** Sacculi of *S. coelicolor* WT, Δ*matAB* and Δ*cslA* were isolated using a protocol based on the method of Glauner[62]. Liquid cultures of 12 h old were resuspended in cold TrisHCl pH 7.0, subsequently boiled in 4% sodium dodecyl sulfate (SDS) for 30 min and washed with milliQ. The sample was enzymatically treated with DNase, RNase and trypsin. Then the sample was again boiled for 30 min in 4% SDS and washed. The sample was pelleted and resuspended in 48% hydrofluoric acid (HF), which is shown to be sufficient to quantitatively remove teichoic acids from the sacculus[63]. After 48 h of HF treatment at 4 °C, the sample was washed multiple rounds and concentrated.

**Cryo-electron tomography.** Before vitrification of the sample, 10 nm colloidal gold beads (Protein A coated, CMC Utrecht) were added to the sacculi suspension as fiducial markers in a 1:20 or 1:50 ratio. Vitrification was performed using a Leica EM GP plunge freezer. The EM-grids were glow-discharged 200 mesh copper grids with an extra thick R2/2 carbon film (Quantifoil Micro Tools). 3.5 μL of the sample was applied to the EM grid and blotted at a temperature of 16 °C and a humidity between 95–99%. Grids were automatically blotted with a blot time of 1 s and plunged into liquid ethane. Samples were mounted on a 626 cryo-specimen holder (Gatan, Pleasanton, CA) and examined using a 120 kV Talos TEM (FEI/ThermoFisher) equipped with Lab6 filament and Cita CCD camera.

Data were collected using a Titan Krios instrument (ThermoFischer Scientific) equipped with a 300 keV electron gun, Volta phase plate and Gatan energy filter with K2 Summit DED (Gatan, Pleasanton, CA). The sample was tilted −60, + 60, and imaged with 2 degrees increment, cumulative exposure of 100 electrons and a pixel size of 4.241 Å (Table 1). Tilt series were acquired using Tomography 4.0 software (ThermoFisher Scientific) with usage of the Volta phase plate (ThermoFisher Scientific) and a defocus set to −0.5 μm. The tomograms were reconstructed by applying a weighted back-projection algorithm with SIRT-like filtering, using IMOD software[64].

***tagO* knockdown via CRISPRi.** The sgRNA scaffold was amplified by PCR using primers SgPermE_F_EBG and SgTermi_R_B on pCRISPR-dCas9[48]. The PCR product was cloned into pHJL401 via *Eco*RI and *Bam*HI to generate construct pGWS1045. Promoter of *gapdh* was amplified by PCR using primer pair Pgapdh_F_(E)B and Pgapdh_R_(H)NdeI on pCRISPomyces-2[65]; *dcas9* was amplified by PCR on pCRISPR-dCas9 using primers Cas9_F+1_(E)NdeI Cas9-Termi_R+4107_XH[48]. PCR products of the *gapdh* promoter and *dcas9* were digested with *Bam*HI–*Nde*I and *Nde*I–*Xba*I, respectively, and then simultaneously cloned into *Bam*HI–*Xba*I digested pGWS1045 to generate construct pGWS1049. The 20 nt spacer sequence was introduced into the sgRNA scaffold by PCR using forward primers TagO_T_F or TagO_NT3_F together with reverse primer SgTermi_R_B. The PCR products were cloned into *Nco*I/*Bam*HI-digested pGWS1049 to generate constructs pGWS1355 and pGWS1358. Subsequently, DNA fragments containing the sgRNA scaffold (with spacer) and P*gapdh*-*dcas9* of constructs pGWS1355 and pGWS1358 were digested with *Eco*RI and *Xba*I and cloned into pSET152 using the same restriction enzymes. Constructs pGWS1362 (targeting template strand of *tagO*, control) and pGWS1365 (targeting non-template strand of *tagO*) were then introduced into *S. coelicolor* M145 via conjugation as described previously[61]. An overview of the constructs and oligonucleotides is presented in Supplementary Table 1.

**Mutanolysin treatment and microscopy.** HF-treated sacculi of *S. coelicolor* were kept in mutanolysin buffer containing mutanolysin. The sample was incubated at 30 °C, and at three time points (0, 15 and 30 min upon start of the treatment) sample was taken for observation with a light microscope and prepared for room temperature TEM. For bright field microscopy, 3 μL of the sample was placed on a glass microscopy slide with cover slip and observed with a Zeiss Axio Lab A1 upright microscope with an Axiocam MRc camera. For TEM, ~20 μL of the sample was applied on a 200 mesh copper with continuous carbon grid and left to dry at room temperature. The sample was observed using a 120 kV Talos TEM (FEI/ThermoFisher) equipped with Lab6 filament and Cita CCD camera.

**Data analysis.** The density plots and thickness measurements of the cryo-ET data were performed with the open-source software ImageJ and (FIJI) plugins[66]. Per sacculus, the approximate middle was set by determining the slice where the sacculus is the broadest. Approximately 50–100 slices around the sacculus was summed to one micrograph, as the sacculi can contain wrinkles or does not always lay flat in one plane. From the summed micrograph, the sacculus was traced using various filters and skeletonization of the micrograph using tools in FIJI. The skeletonized line following the sacculus was used as selection to straighten the sacculus, which was subsequently used for a vertical density/profile plot using the Profile Plot tool in FIJI. For the tip vs. lateral region comparison, a selection of 800 pixels (~679 nm) from the most curved region was designated as the tip region. The density plot of this region was compared to the density plots of adjacent (left and right, if available) lateral regions. The cell-wall thickness per sacculus was determined by subtracting the background signal from the density plot values of the straightened cell wall. The average background pixel value was determined by averaging all pixel values of the regions outside and inside the sacculus selection.

**Statistics and reproducibility.** Statistical analysis of the sacculi thickness was performed using a two-sample Student's *t*-test, using equal variance. Samples were noted as statistically significant at a *p*-value of <0.05. For this analysis, the cell walls of multiple individual sacculi were measured per strain: $n = 9$ sacculi of the WT, WT HF-treated, Δ*matAB* HF-treated, $n = 8$ for Δ*cslA*, $n = 7$ for Δ*cslA* HF-treated, $n = 6$ for Δ*matAB*.

**Reporting summary.** Further information on research design is available in the Nature Research Reporting Summary linked to this article.

## Data availability

The datasets generated during and analyzed during the current study are available from the corresponding author on reasonable request. Source data underlying Figs. 2 and 5 are presented in Supplementary Data 1 and 2.

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

## Acknowledgements

We thank Julio O. Ortiz (Netherlands Center for Electron Nanoscopy (NeCEN), Leiden, The Netherlands) for support on cryo-ET data acquisition with the Volta Phase Plate and critical reading of the manuscript. And we thank J. Willemse (Institute of Biology, Leiden University, Leiden, the Netherlands) for his help with FIJI for data analysis. We thank Dr. G. Muth (Universität Tübingen, Tübingen, Germany) for sharing his experience with teichoic acid mutants in *S. coelicolor* and for sharing unpublished data. This work has been supported by the profile area "Antibiotics" of the Faculty of Sciences of Leiden University, by Grant 731.014.206 from the Netherlands Organization for Scientific Research (NWO) to G.P.v.W. and by iNEXT, PID:2265, funded by the Horizon2020 programme of the European Commission.

## Author contributions

E.U., L.T.v.d.A., D.v.D, G.P.v.W., D.C. and A.B. designed the study; E.U., D.C. and A.B. wrote the manuscript. L.T.v.d.A. and E.U. isolated the sacculi. E.U. prepared the samples for cryo-ET and performed cryo-TEM. E.U. and C.D. performed cryo-ET data collection. E.U. performed the cryo-ET reconstruction and analysis. L.Z. performed knock-out experiments targeting *tagO* and CRISPRi directed knockdown of *tagO*. D.v.D. created the *matAB* and *cslA* knock-out mutants. E.U., L.T.v.d.A., A.B. and D.C. interpreted and discussed the data. All authors discussed the results, read and approved the manuscript.

## Competing interests

The authors declare no competing interests.
