## [Peer Review File · Communications Biology]

Reviewers' comments:

Reviewer #1 (Remarks to the Author):

This manuscript describes a cryo electron tomography analysis of *Streptomyces* cell wall, revealing previously unknown features of the cell wall of the apically growing bacterium. The concept of extracellular glycans being an outer layer of the cell wall is novel, and wall teichoic acid mediating the connections between the conventional inner peptidoglycan layer and the outer glycan cell wall is a significant step forward towards understanding the architectures of bacterial cell walls in general.

The methodology of cryo electron tomography is a robust approach, and the high-quality data displayed in the manuscript generally support the author's proposed model.

However, there are several major concerns that need to be addressed to make the author's claims convincing. First, there is not sufficient evidence to substantiate the author's assumption that 48% HF treatment will quantitatively and selectively remove wall teichoic acids from *Streptomyces* cell wall. The authors employ this treatment condition based on a previously published article. However, the referenced paper reports the analysis of the cell walls from *Streptococcus pneumoniae*, a completely different bacterium.

The authors should provide their own data from *Streptomyces coelicolor* that the same treatment condition removes teichoic acids from the cell wall while other cell wall components remain intact. I am particularly concerned if poly- β -1,6-N-acetylglucosamine and cellulose-like polymer are susceptible to HF. If HF treatment is degrading these glycans, observed patchiness and fraying may have nothing to do with teichoic acids. Such chemical analyses of the cell wall with and without HF treatment will be essential to make the author's data convincing.

My second concern is related to my first concern. Even if HF treatment does remove wall teichoic acids from the *Streptomyces* cell wall, it is still a chemical treatment, which may not be very specific to wall teichoic acids. To make the data more convincing, an alternative approach should be taken. Tunicamycin may be used to block the synthesis, and therefore, the authors test if similar cell wall defects such as cell wall fraying can be observed upon chemical inhibition of wall teichoic acid biosynthesis. Even better, the authors may consider knocking out a gene encoding a wall teichoic acid biosynthetic enzyme. Such additional approaches are critical for the authors to propose the new model of the *Streptomyces* cell wall.

Finally, regarding Figure 5, the authors only included the data on the sacculi of Δ matAB and Δ clsA mutants after HF treatment. Based on author's model, HF treatment should have no effect on the thin cell walls of the mutants. The authors must demonstrate this by including the data on HF-untreated sacculi from Δ matAB and Δ clsA mutants. Such a data, I believe, is an important control experiment.

Additional minor comments are indicated below.

Line 43. Remove "this".

Line 56. There are several more important papers that should be reference here. PMID 30198841, 25049412, 30160378.

Line 68. Consider defining "open-growth morphology" more clearly for non-specialist readers. Would "dispersed cell growth" be better description?

Line 69. It is unclear what "liquid-grown media" means. Do the authors mean "liquid growth media"?

Line 109. As described in the major concerns, this statement is too strong. It remains a formal possibility that another molecule sensitive to HF, rather than TAs, is responsible for the observed phenotype.

Line 118. This sentence has the same problem as above (Line 109), making a strong statement without substantiating data.

Line 138. Revise the awkward and confusing phrase "the dense cultures grown by the parent *S. coelicolor* strain".

Line 156. Missing a period.

Line 219. Indicate the incubation temperature for HF treatment.

Figure 2B. The gray and black lines should be swapped to match the image.

Reviewer #2 (Remarks to the Author):

In this work, authors combined classic chemical treatments, genetically modified strains, and state-of-the-art cryo-electron microscopy approaches to study the cell wall ultrastructure of a tip-grower bacterium. Contrasting the apparent simplicity of the experiments, the authors collected a set of novel insights about the physiological and structural state of different cellular regions of this bacterial cell.

Overall, the paper is well written, pleasantly short, and represent a substantial advance on how we understand the cell wall tip-grown bacteria. There are essential but straightforward revision points in the manuscript. The critical part should be a better description and reconciliation between the data described and the model proposed by the authors.

Specific Comments

1. Even though it is understandable the challenge of collecting large cryo-ET datasets, the authors should be careful while drawing certain conclusions. For example, on lines 96-102, they described the analysis of extracted sacculi from 5 cells and observed a difference between apical and subapical staining in only 3 of them. Provided the low number of analyzed samples, it's a stretch to conclude:

"This data set indicates a structural difference between the apex, where the incorporation of new PG into the existing cell wall takes place, and the relatively older cell wall located subapically".

The authors have done a good work providing a reasonable explanation for the discrepancy (lines 171-172), but they should adapt their assertion by bringing the interpretation closer to the results and tone down their conclusion.

2. Likewise, authors should include in-text the statistics of all 8 analyzed sacculi described in lines 107-109. Fig. 2B shows that at least 1 out of 5 sacculi have similar contrast compared apical and

subapical regions (4th panel from the left). Additionally, authors should include a brief explanation in the legend of Fig. 2A of the reason there is an empty data point (4th panel from left, subapical right).

3. I strongly suggest authors should take advantage of the already quantitated dataset to display the results in a more intuitive way to the reader - optimally as a Fig. 2C. One alternative is to measure the area under each subapical curve, normalize by the apical area and scatter-plot control vs HF treatment.

4. Lines 123-124: Authors describe a 90-minute mutanolysin treatment that is not present in Fig. S3 (only 0, 15 and 30 min). Include data or adapt the text accordingly.

5. Despite the great discussion about the importance of TAs in the lateral cell wall stability, it is confusing to understand how the authors reconcile their model (Fig. 6) with the fact that HF-treatment abolishes the differences between the growing tip and the lateral cell wall (Fig. 2). Fig. 6 shows TAs and extracellular glycan are evenly distributed everywhere around the cell. It seems this is not one of the best interpretations. An interesting explanation is that apical zones actually may lack TAs to make cell wall more plastic for remodeling during growth. That would explain why growing apical areas have less cell-wall staining - as the lack of TAs could destabilize outerglycans that would shed more frequently to the medium. An easy way to test this hypothesis is to perform the measurements done in Fig. 2 on Fig. 5 dataset (mutants w/ and w/o HF) - showing that now sacculi show identical density profiles between apical and subapical regardless of HF treatment.

Reviewer #3 (Remarks to the Author):

The article from Ultee et al describes the structural organization of the cell wall in Actinobacteria. The article is well written and the message is well conveyed. I am not an expert in microbiology, therefore I will not comment on the interest of the findings in the field, but I did find it interesting as an outsider.

I am providing a technical review on the tomography, and I have 2 comments which can be easily answered by the authors, but that are important:

- throughout the figures, it is not clear whether the panels (e.g. Fig2, Fig1E-F, etc.) are XY slices or if they are from a different axis. the figure legends are limited and in my opinion, should be more descriptive.

- In many of the figures the signal appears stretched uniformly in one direction (e.g. Fig 2, Fig 3F) this can be partly imputed to the missing wedge in association with a specific orientation of the slice. In order to assess that the interpretation is correct, it is imperative to see the tomograms. I suggest seeing a selection of representative tomograms (possibly matching the features shown in the article through upload on EMPIAR/EMBD (an action that should be anyway a normal procedure nowadays for this type of article).

I will comment again after having seen the tomograms. If the alignment is correct there will be no changes requested on this aspect and I would recommend the publication.

Reviewer #1

We would like to thank the reviewers for their insightful and constructive comments and suggestions, and revised the manuscript accordingly. We will respond to each reviewer comment below.

First, there is not sufficient evidence to substantiate the author's assumption that 48% HF treatment will quantitatively and selectively remove wall teichoic acids from Streptomyces cell wall. The authors employ this treatment condition based on a previously published article. However, the referenced paper reports the analysis of the cell walls from Streptococcus pneumoniae, a completely different bacterium.

My second concern is related to my first concern. Even if HF treatment does remove wall teichoic acids from the Streptomyces cell wall, it is still a chemical treatment, which may not be very specific to wall teichoic acids. To make the data more convincing, an alternative approach should be taken. Tunicamycin may be used to block the synthesis, and therefore, the authors test if similar cell wall defects such as cell wall fraying can be observed upon chemical inhibition of wall teichoic acid biosynthesis. Even better, the authors may consider knocking out a gene encoding a wall teichoic acid biosynthetic enzyme. Such additional approaches are critical for the authors to propose the new model of the Streptomyces cell wall.

We understand the Reviewer's concern on the use of HF to remove teichoic acids from the Streptomyces cell wall. To address this, we decided to attempt generating a mutant lacking teichoic acids. As a target, we selected *tagO*, since the TagO protein catalyses the first step in wall teichoic acid. Therefore, this mutant would lack wall teichoic acids. However, we were unable to generate a viable null mutant. This result agrees with research group of Günther Muth at Tübingen University, as they were also unable to create a *tagO* mutant. We then decided to use a dCas9 interference approach, which allowed us to successfully generate a *tagO* knockdown mutant strain. However, this strain had a significant germination and growth defect, which affected sacculi isolation and further cryo-ET assessment of the hyphal tips. We have included these results to the manuscript.

The authors should provide their own data from Streptomyces coelicolor that the same treatment condition removes teichoic acids from the cell wall while other cell wall components remain intact. I am particularly concerned if poly- β -1,6-N-acetylglucosamine and cellulose-like polymer are susceptible to HF. If HF treatment is degrading these glycans, observed patchiness and fraying may have nothing to do with teichoic acids. Such chemical analyses of the cell wall with and without HF treatment will be essential to make the author's data convincing.

The Reviewer suggested chemical analyses of the cell wall with and without HF treatment to ensure that the extracellular polymers, poly- β -1,6-N-acetylglucosamine (PNAG) and the cellulose-like polymer, are not degraded by the HF-treatment. It is unlikely the extracellular polymers are affected since HF is a very weak acid and its major activity is breaking phosphodiester bonds due to its fluoride. To address the concerns expressed by the Reviewer, we utilized (immuno)fluorescence microscopy targeting the extracellular glycans.

First, we investigated the CslA-produced cellulose-like polymer with calcofluorwhite. Calcofluorwhite binds specifically to β -(1,4)linked glycans, such as cellulose and the cellulose-like polymer produced by CslA (Xu, Chater, Deng, & Tao, 2008). but not the PNAG produced by the MatAB proteins (van Dissel et al., 2018). Our results however show that CFW is a difficult stain to work with, as it binds to

all isolated sacculi material and not merely to the hyphal tips as previously shown. Additionally, it binds to isolated sacculi of the *csIA* null mutant.

Second, we used immuno-fluorescence labelling to detect the MatAB-produced PNAG polymers as previously described (van Dissel et al., 2018). Again, this gave inconclusive results as the immunofluorescent labelling targeted the sacculi of the *matAB* null mutant, both with and without HF treatment.

As a result of the inconclusive outcomes of these experiments, we did not include these experimental results in the manuscript. We provide the outcome of these experiments as an attachment to this response for the reviewer (see below).

Finally, regarding Figure 5, the authors only included the data on the sacculi of $\Delta matAB$ and $\Delta csIA$ mutants after HF treatment. Based on author's model, HF treatment should have no effect on the thin cell walls of the mutants. The authors must demonstrate this by including the data on HF-untreated sacculi from $\Delta matAB$ and $\Delta csIA$ mutants. Such a data, I believe, is an important control experiment.

We wish to thank the reviewer for this suggestion. We have acquired cryo-ET data on the non HF-treated sacculi of the *matAB* and *csIA* null mutants, which has given more insight into the cell wall and allowed us to refine our model. We have added these new results to the manuscript (See updated figure 5). Following the outcome of these control experiments, we have adapted our model (figure 6) and description thereof.

We have adjusted the text in the results section as follows:

'The WT sacculi had an average thickness of 30.16 nm (\pm 0.88, n=9). The thickness of the HF-treated WT sacculi showed an average thickness of 31.19 (\pm 2.37, n=9) and a higher variance, presumably caused by the patches and double layers in this sample group. In contrast, the non-treated sacculi of the $\Delta matAB$ and $\Delta csIA$ strains showed an average thickness of 35.91 (\pm 1.73, n=6) and 34.35 nm (\pm 1.71, n=8), respectively. Whereas HF-treatment leads to a thinner cell wall of 23.75 (\pm 1.72, n=9) for $\Delta matAB$ and 22.78 nm (\pm 1.49, n=7) for the $\Delta csIA$ mutant strain. The values of the WT significantly differ from the cell wall thickness of the non-treated $\Delta matAB$ and $\Delta csIA$ sacculi ($p < 0.05$) and the HF-treated $\Delta matAB$ and $\Delta csIA$ sacculi ($p < 0.005$) when analyzed using the Student's *t* test.

The remarkably thicker cell wall and appearance of cell wall lamella in sacculi of both the *matAB* and *csIA* null mutants suggests that the absence of extracellular polymers negatively influences the integrity and compact nature of the cell wall. Additionally, the significantly thinner cell wall of both the *matAB* and *csIA* null mutants indicates that the extracellular glycans produced by the MatAB proteins and CslA comprise a considerable part of the cell wall itself. Moreover, the absence of patches and lack of double layers in these mutants imply that these patches are composed to a large degree of these glycans, and that these glycans are thus an integral part of the cell wall.'

And the text in the discussion:

' The structural integrity of the cell wall was additionally affected in sacculi of *csIA* and *matAB* null mutants, lacking the extracellular polymers. The absence of extracellular polymers led to sacculi with a thicker appearance, with sporadically revealing lamella as seen in HF-treated WT sacculi. In contrast, the HF-treated sacculi of *csIA* and *matAB* null mutants lacked any observable lamellae, strongly suggesting that lamellae are composed of - or depend on - the extracellular matrix synthesized by the CslA and MatAB proteins. The results presented in this study warrant a revisited model for the cell wall of the polar growing bacterium *S. coelicolor* (Figure 6)'

Additional minor comments were suggested by Reviewer #1.

Line 43. Remove “this”.

The text has been adjusted as suggested.

Line 56. There are several more important papers that should be reference here. PMID 30198841, 25049412, 30160378.

Thank you. These references have been added to the manuscript.

Line 68. Consider defining “open-growth morphology” more clearly for non-specialist readers. Would “dispersed cell growth” be better description?

Thank you. We have changed the “open-growth morphology” to “dispersed cell growth”.

Line 69. It is unclear what “liquid-grown media” means. Do the authors mean “liquid growth media”?

We have changed “liquid-grown media” to “liquid growth media” as suggested.

Line 109. As described in the major concerns, this statement is too strong. It remains a formal possibility that another molecule sensitive to HF, rather than TAs, is responsible for the observed phenotype.

We have adjusted the text as suggested:

The data presented here indicates that HF-treatment affects the cell wall density differences observed between the apical and subapical regions, therefore it could be hypothesized that TAs might be involved in the cell wall integrity.

Line 118. This sentence has the same problem as above (Line 109), making a strong statement without substantiating data.

Thank you. We have adjusted this statement as well:

‘Taken together, it could be suggested that teichoic acids play a role in the structural integrity of the *S. coelicolor* cell wall’.

Line 138. Revise the awkward and confusing phrase “the dense cultures grown by the parent *S. coelicolor* strain”.

Done

Line 156. Missing a period.

Done

Line 219. Indicate the incubation temperature for HF treatment.

The incubation temperature for the HF treatment has been added to the Methods section.

Figure 2B. The gray and black lines should be swapped to match the image.

The suggestion made by the Reviewer was not entirely clear. We have revised Figure 2B to hopefully improve the clarity of the figure.

Reviewer #2

1. Even though it is understandable the challenge of collecting large cryo-ET datasets, the authors should be careful while drawing certain conclusions. For example, on lines 96-102, they described the analysis of extracted sacculi from 5 cells and observed a difference between apical and subapical staining in only 3 of them. Provided the low number of analyzed samples, it's a stretch to conclude:

"This data set indicates a structural difference between the apex, where the incorporation of new PG into the existing cell wall takes place, and the relatively older cell wall located subapically". The authors have done a good work providing a reasonable explanation for the discrepancy (lines 171-172), but they should adapt their assertion by bringing the interpretation closer to the results and tone down their conclusion.

We thank the Reviewer for his comments on the interpretations and conclusions drawn from our data sets. We have adapted the manuscript and we have toned down our statements as suggested. Furthermore, we have added more experimental data (suggested by reviewer 1) that allowed us to refine our interpretations and suggested cell wall model.

2. Likewise, authors should include in-text the statistics of all 8 analyzed sacculi described in lines 107-109. Fig. 2B shows that at least 1 out of 5 sacculi have similar contrast compared apical and subapical regions (4th panel from the left). Additionally, authors should include a brief explanation in the legend of Fig. 2A of the reason there is an empty data point (4th panel from left, subapical right).

Thank you for these suggestions. Based also on comments from Reviewer 3, we have added an additional graph to Figure 2, on which statistical analysis was performed. In line with the previously mentioned challenge of collecting large cryo-ET datasets, the low number of samples do not seem appropriate for statistical analysis. Hence we adapted the text to fit the observations made on the differences in the apical and subapical regions. We aimed to refrain from strong conclusions based on these data points (figure 2).

Additionally, as suggested, we have added a brief explanation in the legend of Figure 2 why there is an empty data point (panel $\Delta matAB$, right subapical region):

'Not all sacculi contain a left and right subapical region in the 21 tomogram field of view, as depicted by the missing panel in the $\Delta matAB$.'

3. I strongly suggest authors should take advantage of the already quantitated dataset to display the results in a more intuitive way to the reader - optimally as a Fig. 2C. One alternative is to measure the area under each subapical curve, normalize by the apical area and scatter-plot control vs HF treatment.

As suggested by the Reviewer, we have used our density plots to further determine the area under each subapical curve, normalize by the apical area and plot the outcome as an additional graph in Figure 2.

4. Lines 123-124: Authors describe a 90-minute mutanolysin treatment that is not present in Fig. S3 (only 0, 15 and 30 min). Include data or adapt the text accordingly.

Thank you for pointing this out. We have adapted the text describing a 90-minute mutanolysin treatment to fit the data presented in former Figure S3, now Figure S4.

5. Despite the great discussion about the importance of TAs in the lateral cell wall stability, it is confusing to understand how the authors reconcile their model (Fig. 6) with the fact that HF-treatment abolishes the differences between the growing tip and the lateral cell wall (Fig. 2). Fig. 6 shows TAs and extracellular glycan are evenly distributed everywhere around the cell. It seems this is not one of the best interpretations. An interesting explanation is that apical zones actually may lack TAs to make cell wall more plastic for remodeling during growth. That would explain why growing apical areas have less cell-wall staining - as the lack of TAs could destabilize outerglycans that would shed more frequently to the medium. An easy way to test this hypothesis is to perform the measurements done in Fig. 2 on Fig. 5 dataset (mutants w/ and w/o HF) - showing that now sacculi show identical density profiles between apical and subapical regardless of HF treatment.

Thank you for this suggestion. We have applied the measurements performed in Figure 2 (density difference between apical and subapical regions) to the datasets of the cryo-ET on $\Delta matAB$ and $\Delta cslA$ sacculi, both with and without HF-treatment. This has been incorporated into the manuscript.

Reviewer #3

- throughout the figures, it is not clear whether the panels (e.g. Fig2, Fig1E-F, etc.) are XY slices or if they are from a different axis. the figure legends are limited and in my opinion, should be more descriptive.

We thank the Reviewer for this insight, and we have adapted the legends holding panels (Figure 1E-F, Figure 2, Figure 5) to point out the fact that these concern XY slices of the tomograms.

- In many of the figures the signal appears stretched uniformly in one direction (e.g. Fig 2, Fig 3F) this can be partly imputed to the missing wedge in association with a specific orientation of the slice. In order to assess that the interpretation is correct, it is imperative to see the tomograms. I suggest seeing a selection of representative tomograms (possibly matching the features shown in

the article through upload on EMPIAR/EMBD (an action that should be anyway a normal procedure nowadays for this type of article).

I will comment again after having seen the tomograms. If the alignment is correct there will be no changes requested on this aspect and I would recommend the publication.

As suggested by the Reviewer, we will upload the raw tilt series of the tomograms represented in the manuscript on EMPIAR. As uploading the data onto this database could be subject to a delay, we have provided a link to a OneDrive folder from which the data can be downloaded and assessed until they are available on EMPIAR.

Link:

https://leidenuniv1-my.sharepoint.com/:f:/g/personal/ultee_vuw_leidenuniv_nl/EgKh7XybbbNPtWzYoxoVPZAB9_UONIwK2JS_6PbaxSNKhw?e=Ep273f

Sacculi stained with CFW

S. coelicolor sacculi of the WT, $\Delta cs/A$ and $\Delta matAB$ strains were stained with calcofluorwhite.

Sacculi stained with immuno-labelling

S. coelicolor sacculi of the WT, $\Delta csIA$ and $\Delta matAB$ strains were stained with PNAG antibodies, according to methods described by Van Dissel et al. 2018, *Microbial Cell* (doi:10.15698/mic2018.06.635)

REVIEWERS' COMMENTS:

Reviewer #1 (Remarks to the Author):

None. All concerns addressed.

Reviewer #2 (Remarks to the Author):

I am satisfied with the corrections and responses given by the authors. It is also commendable to their effort to generate a tagO knockdown strain to test their model.

However, I suggest the authors include a better description of how the dCas9 induction experiments were done (either in methods or along with the results section). Knowing the induction details (like inductor concentration and induction period) may be important to understand the reason for the comparable results between wild type and TagO-depleted cells in the future.

Reviewer #3 (Remarks to the Author):

I am satisfied with the data and with the response of the authors.
I apologize for the delayed response, I have been unwell.